# Impact of Atorvastatin on Skeletal Muscle Mitochondrial Activity, Locomotion and Axonal Excitability—Evidence from *ApoE^-/-^* Mice

**DOI:** 10.3390/ijms23105415

**Published:** 2022-05-12

**Authors:** Chiara Macchi, Veronica Bonalume, Maria Francesca Greco, Marta Mozzo, Valentina Melfi, Cesare R. Sirtori, Valerio Magnaghi, Alberto Corsini, Massimiliano Ruscica

**Affiliations:** Dipartimento di Scienze Farmacologiche e Biomolecolari, Università degli Studi di Milano, 20133 Milan, Italy; chiara.macchi@unimi.it (C.M.); veronica.bonalume@unimi.it (V.B.); mariafrancesca.greco@unimi.it (M.F.G.); mozzomarta@gmail.com (M.M.); valentina.melfi@unimi.it (V.M.); cesare.sirtori@icloud.com (C.R.S.); valerio.magnaghi@unimi.it (V.M.); alberto.corsini@unimi.it (A.C.)

**Keywords:** statins, atorvastatin, mitochondria, locomotion, muscles, grip test

## Abstract

The cardiovascular benefit of statins is well established. However, only 20% of high-risk patients remain adequately adherent after 5 years of treatment. Among reasons for discontinuation, statin associated-muscle pain symptoms are the most prevalent. Aim of the present study was to evaluate the impact of high dose atorvastatin on skeletal muscle mitochondrial activity, aerobic and anaerobic exercise, and axonal excitability in a murine model of atherosclerosis. *ApoE^-/-^* mice were fed 12 weeks a high-fat high-cholesterol diet alone or containing atorvastatin (40 mg/Kg/day). Outcomes were the evaluation of muscle mitochondrial functionality, locomotion, grip test, and axonal excitability (compound action potential recording analysis of Aα motor propioceptive, Aβ mechanoceptive and C nociceptive fibres). Atorvastatin led to a reduction in muscle mitochondrial biogenesis and mitochondrial ATP production. It did not affect muscular strength but led to a time-dependent motor impairment. Atorvastatin altered the responsiveness of mechanoceptive and nociceptive fibres, respectively, the Aβ and C fibres. These findings point out to a mild sensitization on mechanical, tactile and pain sensitivity. In conclusion, although the prevalence of muscular side effects from statins may be overestimated, understanding of the underlying mechanisms can help improve the therapeutic approach and reassure adherence in patients needing-to-be-treated.

## 1. Introduction

Statins are highly effective drugs that lower plasma low-density lipoprotein (LDL) cholesterol (LDL-C), on average by 30% to 50% [1] by competitively inhibiting hydroxymethylglutaryl-coenzyme A reductase, the rate-limiting enzyme in the cholesterol biosynthesis pathway [2]. Thus, statins are recommended as the first choice for the management of hypercholesterolaemia and combined hyperlipidaemia [3,4]. Statins act mainly in the liver, where the expression of LDL-receptors is increased due to the decreased hepatocyte cholesterol [5], thus raising the clearance of LDL-C from the bloodstream [6].

Although genetic, interventional, and observational studies have undoubtedly proven a statin-driven atherosclerotic cardiovascular disease benefit [7], 30–50% of patients stop their statin after 6 months, and only 20% of high-risk patients remain adherent after 5 years [8]. Among reasons for discontinuation, statin associated-muscle symptoms are the most prevalent adverse effects [9]. Statin associated-muscle symptoms may appear as myalgia, myopathy, myositis with or without creatine kinase elevation or in the most severe cases as rhabdomyolysis [10]. In this context, several lines of evidence suggest that statin associated-muscle symptoms may be due to an impairment of mitochondrial membrane potential, maximal oxygen uptake and ATP levels [11]. Mitochondrial dysfunction is more pronounced in statin users with statin associated-muscle symptoms compared with asymptomatic statin users [12]. Interestingly, in some cases, despite normal blood biochemistry (normal creatine kinase) it is possible that muscle biopsies show mitochondrial dysfunction [13]. There is, further, no apparent difference in muscle damage after aerobic (locomotion type) exercise, or anaerobic (strength) exercise [14] with very few controlled investigations and essentially no animal studies addressing this variable.

Another aspect worthy of consideration is the impact of statin use on the rate of peripheral neuropathy, a common clinical problem affecting at least 8% of the population >55 years of age [15], even greater if considering only peripheral nerve injuries. This condition, characterized by dysfunction and damage to peripheral nerves, appears to be dependent on numerous factors and occurs typically with lipophilic statins. Peripheral neuropathy may affect different fibres due to different aetiologies and can be classified according to type of fibres affected and pattern of injury [16], as in the Guillain-Barrè syndrome found upon initiation of statin therapy and characteristically associated with altered nerve conduction [17]. Although observational studies suggest a possible association between statin use (prior or current) and newly diagnosed peripheral neuropathy [18,19], primary peripheral neuropathy is substantially more common and according to some, even more frequent than muscle injury [18], leading to one excess case of idiopathic peripheral neuropathy for every 2200 (880 to 7300) person-years of statin use [20]. However, caution has been expressed on the validity of these findings [21]. Indeed, there is no support for a causal relationship either in randomized controlled trials [22] or from an aetiological point of view. Neuronal cholesterol biosynthesis impairment does in fact occur with a lipophilic statin such as simvastatin, with reduced synapse density and impaired synaptic vesicle release even when lipoproteins or geranylgeraniol are added to the system [23]. It appears therefore of interest to evaluate neuropathy in standardized conditions in an appropriate animal model.

Thus, aim of the present study was to evaluate in a mouse model of atherosclerosis, namely, *ApoE^-/-^* mice fed a high-fat high-cholesterol diet (HFHC) [24,25], the impact of 12-week treatment with a high dose atorvastatin (40 mg/Kg/day) on skeletal muscle mitochondrial activity, locomotion, grip strength and axonal excitability.

## 2. Results

### 2.1. Impact of Atorvastatin on Lipid and Glycaemic Profiles

The effectiveness of a short-term treatment (1 week) with atorvastatin (40 mg/Kg/day) was assessed on circulating lipid levels in *ApoE^-/-^* mice fed a HFHC (control group) and a HFHC plus atorvastatin (Figure 1A). In the group given atorvastatin, total cholesterol (TC), high-density lipoprotein cholesterol (HDL-C), non-HDL-C, and triglycerides (TG) were reduced by 36% (*p* = 0.056), 18% (*p* = 0.31), 38% (*p* = 0.032), and 15% (*p* = 0.69), respectively. The robust drop in lipids was confirmed after 12 weeks of treatment, namely, −30.9% (TC; *p* < 0.0001), −20.8% (HDL-C; *p* = 0.0014), −31.5% (non-HDL-C; *p* < 0.0001), and −24.2% (TG; *p* = 0.021) (Figure 1B). Concerning glycaemia, no between-group differences were found in both basal levels and after a glucose tolerance test (Figure 1C).

### 2.2. Impact of Atorvastatin on Body Weight and Organs’ Weight

Compared with the control group (mice fed HFHC), those receiving atorvastatin (40 mg/Kg/day) showed a reduction in body weight. This phenotype was already evident after 6 weeks and was independent of food intake (Figure 2A,B). Concerning organs, after 12 weeks, the group given atorvastatin had a significant reduction in the weight of visceral adipose tissue (VAT; −19.3%) without changes in liver, spleen, kidney, and skeletal muscles (Figure 2C).

### 2.3. Impact of Atorvastatin on Mitochondrial Biogenesis and Functionality

First, we assessed the impact of atorvastatin on mitobiogenesis in quadriceps. In mice given atorvastatin, gene expression of mitochondrial transcription factor (*Tfam*) was significantly reduced (Figure 3A), whereas the phosphorylated protein levels of serine/threonine kinase AMP-activated protein kinase (AMPK) showed a trend toward reduction (*p* = 0.0696; Figure 3B). The protein expression of all replicates is reported in Appendix A. Next step was the evaluation of mitochondrial respiration conducted on fresh mitochondria isolated from skeletal muscles (a pool of tibialis anterior, extensor digitorum longus, soleus, gastrocnemius, quadriceps, and biceps brachii), immediately after sacrifice. *ApoE^-/-^* mice given atorvastatin had a significant reduction of basal respiration (−22.9%; *p* = 0.004), an effect maintained when the uncoupler of mitochondrial oxidative phosphorylation FCCP was added (−21.7%; *p* = 0.024). ATP production was also reduced (−21%; *p* = 0.049; Figure 3C). Conversely, after a short-period of treatment (1 week), atorvastatin did not affect mitochondrial respiration (data not shown).

To further explore changes found by energetic analysis, the mitochondrial dynamicity was investigated. As shown in Figure 4, while no changes were found in the protein expression of dynamin related protein 1 (DRP-1; Figure 4A), involved in mitochondrial fission, optic atrophy 1 (OPA1), involved in mitochondrial fusion, was significantly increased in the skeletal muscles of mice given atorvastatin (Figure 4B). For both DRP-1 and OPA-1, the protein expression of all replicates is reported in Appendix A.

### 2.4. Impact of Atorvastatin on Skeletal Muscle Strength and Locomotion

To evaluate the impact of treatment with atorvastatin on muscular performance, *ApoE^-/-^* mice underwent a set of behavioural tests: grip and walking test and rotarod analysis. The grip analysis showed that 12-week administration of atorvastatin did not impact strength performance compared to mice fed only HFHC (n.s., Figure 5A). However, when both experimental groups were analysed individually, a significant time-dependent decrease in grip test was found after 12 weeks compared to the analysis at week 6 (Figure 5A). The percentage decrement was comparable between mice fed a HFHC (−21.33%; *p* < 0.001) and a HFHC plus atorvastatin (−24.1%; *p* < 0.001). This supports the conclusion that chronic treatment with atorvastatin did not affect muscular strength. Regarding walking analysis, we focused our attention on the stride length (SL) and stride width (SW), that changed time-dependently (Appendix A). Although no between-group differences in SL and SW were found after 12 weeks (n.s., Figure 5B,C), within-group analysis highlighted significant changes (Figure 5B,C). Specifically, mice fed only HFHC showed a +8.1% rise in SL (*p* < 0.01; Figure 5B) during a period of 12 weeks (T12) compared to a shorter period, namely, 6 weeks (T6). Conversely, mice given atorvastatin showed a consistent decrement after 12 weeks (−8.01%; *p* < 0.001; Figure 5B), suggesting a time-dependent motor impairment (shown in the representative footprints, Appendix A). In addition, mice fed only HFHC showed a decrement in SW during growth (*p* < 0.001; Figure 5C), whereas mice given atorvastatin did not show any variation at the same time-frame (n.s.; Figure 5C). The rotarod analysis showed that the latency of fall, which assesses sensory-motor function and central coordination, was unaltered after 12 weeks with no between-group differences (n.s., Figure 5D). However, when analyses were stratified within groups, similarly to SL, mice fed the HFHC showed a significant time-dependent improvement in fall latency (+4.67%; *p* < 0.001) at T12 (Figure 5D). Instead, atorvastatin did not change fall latency over time (Figure 5D).

### 2.5. Impact of Atorvastatin on Nerve Conduction

To evaluate the impact of atorvastatin on nerve conduction, *ApoE^-/-^* mice underwent compound action potential (CAP) recording analysis of peripheral nerve fibres: Aα (motor propioceptive fibres), Aβ (mechanoceptive fibres) and C (nociceptive fibres). In particular, Aβ fibres were selectively recorded from the sural nerve, a pure sensory nerve. After 12 weeks of treatment with atorvastatin, we observed a significant decrement in the stimulus response slope (Figure 6A,B), indicating a lower electrical threshold and a putative increase in mechanic sensitivity. The electrical strength-duration properties (Figure 6C,D), i.e., the time constant (*τ*_SD_), did not change between groups. Then, further tests were used to evaluate the possible modification in axonal membrane potential of Aβ fibres. The first electric protocol, i.e., the threshold electrotonus, showed unaltered waveforms after chronic treatment (Figure 6E), indicating no differences in nodal and internodal conduction. The absence of axonal membrane potential alterations was confirmed also by the second protocol which measures the current-threshold I/V (Figure 6F). Furthermore, the action potential refractoriness in Aβ axons did not change between groups (Figure 6G).

Collectively, the decrease in mechano-receptor electrical threshold found in Aβ fibres, in mice given atorvastatin, was paralleled by the modulation of C nociceptors. These fibres showed a decreased electrical threshold with a left shift in the stimulus response curve (red curve in Figure 7A). The decrement in EC_50_ was 12.8 when HFHC plus atorvastatin and HFHC were compared (*p* < 0.01; Figure 7B). The median EC_50_ was 79.72 (coefficient of variation 12.4 %) for HFHC and 66.9 (coefficient of variation 12.38%) for HFHC plus atorvastatin. This lower electrical threshold may indicate a higher nociception.

The next step was to look at Aα fibres by recording the activity of peroneal nerves, a mixed motor-sensory nerve. Given that in Aβ and C fibres we found major alterations only in axonal electrical threshold, we applied a similar protocol also to Aα fibres. We found no changes in stimulus-response slope (Figure 8A,B), indicative of electrical threshold, whereas *τ*_SD_ parameter, indicative of the electrical strength-duration, was modulated (Figure 8C). Indeed, the *τ*_SD_ was significantly higher in HFHC (*p* < 0.05) vs. those given atorvastatin (Figure 8D).

## 3. Discussion

The main findings of the present study, dealing with the impact of atorvastatin on skeletal muscle mitochondrial activity, exercise, and axonal excitability, were two-fold: first, atorvastatin led to a reduction in mitochondrial ATP production, second, atorvastatin altered the responsiveness of mechanoceptive and nociceptive fibres, respectively, the Aβ and C fibres.

Muscle complaints from statins are of common occurrence in treated patients. While psychological causes have been called into question (nocebo or drucebo effects) [26], their high frequency in athletes has indicated that muscle energy production may be in play [27]. Statin discontinuation, mostly due to intolerance, raises the risk of cardiovascular disease and mortality [28]. Several lines of evidence suggest that statin associated-muscle symptoms may be due to an impairment of mitochondrial membrane potential, maximal oxygen uptake, and ATP levels [12,29].

Mitochondrial dysfunction has become a major factor in a number of, particularly, metabolic abnormalities [30]. The crucial role of mitochondria is well explained by their pillar role in a myriad of processes supporting cellular function, transferring energy of different sources to ATP, mainly produced from mitochondrial fatty acid oxidation, to a lesser extend from glucose, branched chain amino acids and others [31]. Mitochondrial dysfunction, in addition to being a feature of genetic metabolic diseases, may be responsible for drug induced disorders and also for abnormalities occurring in, e.g., the Coronavirus disease (COVID)-19 infection, characterized by dysregulated oxidative phosphorylation [32], or in cancer, where elevated metabolic pathways are needed to support metabolism with raised lactate release and ATP production, especially in tumours with more severe Krebs cycle abnormalities [33]. The case of statins offers an important insight into the potential alteration of the mitochondrial pathways leading to dysfunction. This may occur in mechanistic fashion altering, e.g., the mitochondrial respiratory chain [34] or in clinical conditions such as myocardial ischaemia-reperfusion injury [35] where this mechanism may act as a preconditioner, potentially leading to cardioprotective effects with reduced infarct size. In our model, we found that basal respiration of freshly isolated muscle mitochondria was reduced by 21.2%, along with a decrement in ATP production. In our model, the impaired mitochondrial ATP production was coupled with the reduction in mitochondrial biogenesis as assessed by a lower expression of *Tfam* and pAMPK compared to mice assigned to HFHC. Concerning TFAM, different conclusions have been reached depending on the statin used, namely, in rats given simvastatin there were no changes in the expression of TFAM [36], whereas a decrement was found in rats given atorvastatin [37]. Relative to the expression of muscular AMPK, although the reduction we saw was not significant enough to reach statistical significance (*p* = 0.069), it has to be considered that one ancestral function of AMPK includes control of mitochondrial number through stimulation of mitochondrial biogenesis, regulation of the shape of the mitochondrial network in cells, and mitochondrial quality control through regulation of autophagy and mitophagy [38,39]. Mitochondrial structure is maintained through a balance between the activities of several fusion and fission machineries. While no significant change was noted in DRP-1, involved in fission, there was a significant rise in the protein expression of OPA1, essential for mitochondrial fusion, after atorvastatin. Relative to DRP-1, our data are in line with a previous study in which the hypothesis that a 6-week regimen of simvastatin would attenuate skeletal muscle adaptation to low-intensity exercise was tested [40].

OPA1 in muscles triggers a cascade of signalling that reverberates to the whole body, affecting general metabolism and aging [41]. Thus, changes in muscle mitochondrial functionality upon atorvastatin are consistent with the idea that muscle mitochondria are able, at least initially, to adapt to a reduced ATP production and a decreased mitobiogenesis. In support of this hypothesis, OPA1 overexpression blunts damage of highly metabolically active organs in response to apoptotic, necrotic, and atrophic stimuli [42], as well as ameliorates motor performance in a model of myopathy caused by muscle-specific deletion of the mitochondrial subunit cytochrome c oxidase 15 [43].

However, these findings have to be interpreted within the context of potential limitations. First, mitochondrial biogenesis may be regulated by reactive oxygen species production and oxidative stress [44] and this can explain why some tissues and cells, in particular skeletal muscle fibers, may be most susceptible. Second, statins may inhibit the electron transport chain complexes, thus corroborating the clinical finding indicative of mitochondrial dysfunction leading to impaired fatty acid oxidation [13]. Third, calcium and chloride mediated muscle activities should be mentioned. Patients with statin muscle symptoms have reduced frequency of spontaneous Ca^++^ sparks, increased Ca^++^ spark amplitude [45], and an abnormal cell permeability. As seldom reported, non-energy related cellular changes have been described to potentially raise muscle contractility and consequent toxicity. Among these, chloride channel suppression by statins, and also by fibrates, leads to enhanced contractility, a potentially important factor in inducing muscle pain and toxicity [46,47].

An original part of the present report was the testing of muscle function in an aerobic type of exercise, as exemplified by unforced locomotion, and an anaerobic test as assessed by muscle strength (grip test). Limited information on this topic has been provided by clinical studies, atorvastatin apparently having a modest influence on both exercise types [48]. In obese/metabolic syndrome patients treated with simvastatin, a reduced response to aerobic exercise training was noted with a lower increment in oxygen uptake and 4.2% reduction in citrate synthase activity [49].

The painful syndrome occurring at the limb level of patients on statins has been attributed by some to polyneuropathy. This syndrome is of relatively frequent occurrence in severe metabolic disorders such as diabetes, chronic alcohol abuse and genetic disorders, incidence increasing with age [50]. Several reports have suggested that statin use may be associated with axonal neuropathy [51]. In an early cross-sectional study the association between statin use and peripheral neuropathy was modest [52]; conversely, the association between statin use and polyneuropathy risk in a Danish population was found to be negative [53]. In our model with atorvastatin, we found an increase in Aβ electrical sensitivity and decrease in C fibre electrical threshold, a possible mirror of a higher tactile, thermal and nociceptive perception. Relative to changes found in Aα fibres, these can be ascribed either to Aα motor efferent fibres or to Aα propioceptive afferents. Although the observed effects of atorvastatin on Aα fibres (i.e., modulation of electrical strength-duration time constant) might be indicative of a preservation of motor conduction, two points must be raised. First, considering that the peroneal Aα fibre includes two subpopulations, the efferent motor fibres and the propioceptive afferent fibres, the effect observed might be consequent to a net effect of one population on the other. Second, the between-group difference in τ_SD_ could be mostly ascribed to an increase in tissue lipid deposition in mice fed a high-cholesterol diet. We speculate this effect may be consequent to an alteration in axonal structure, secondary to changes in lipid membrane composition, following the diet. Taken together, the outcomes from peripheral nerve recordings, point out to a mild sensitization driven by atorvastatin on mechanical and pain sensitivity, without clear effects on propioceptive and motor conductance.

## 4. Materials and Methods

### 4.1. Animals and Dietary Regimen

Thirty 5-week-old female *ApoE^-/-^* mice were bought from Charles River (Milan, Italy). After 1 week of acclimatation, mice were fed either a HFHC (*n* = 15) or HFHC containing atorvastatin (*n* = 15) (350 mg/kg of diet per 4 g per daily food intake; Mucedola, Milan, Italy) [54] for further 12 weeks [55]. Composition of the diet is reported in Appendix A. After 1 week of diet regimen, 5 animals per group were sacrificed for the initial biochemical evaluations (see Section 4.3 and Section 4.4; Figure 9).

### 4.2. Chemicals

Atorvastatin calcium 1 salt trihydrate ≥98.0% (code: TCIAA2476-5G; WVR, Milan, Italy). Adenosine diphosphate (ADP), Oligomycin, Carbonyl cyanide-4 (trifluoromethoxy) phenylhydrazone (FCCP), Rotenone, Antimycin A were all purchased from Sigma Aldrich (Milan, Italy).

### 4.3. Biochemical Evaluations

At sacrifice, blood was collected in Eppendorf tubes and centrifuged for 15 min at 2000 rpm at 4 °C. Serum was collected and immediately stored at −80 °C for the evaluation of TC, HDL-C and TG (Cobas, Roche). Non-HDL-C was calculated as TC minus HDL-C.

### 4.4. Glucose Tolerance Test

Mice were fasted overnight (12 h), weighed and blood glucose measured by snipping the tail and using a glucose meter (ONE-TOUCH Ultra) before and after intraperitoneal 1g/kg injection of glucose solution. At the end of the test all the tail wounds were cauterized, and mice were provided with food. The area under the curve (AUC) was calculated for glucose clearance following glucose tolerance test.

### 4.5. Mitochondria Isolation and Mitochondrial Respiration Analysis

To assess the impact of the dietary regimens on skeletal muscle mitochondrial functionality, a Mito Stress Test (Agilent Technologies, Santa Clara, CA, USA) was performed. Fresh mitochondria were isolated from the following skeletal muscles: tibialis anterior, extensor digitorum longus, soleus, gastrocnemius, quadriceps, and biceps brachii. They were weighed, placed in a cold basic medium, minced, and homogenized with a glass/Teflon Potter Elvehjem tissue grinder in a homogenization medium added with Proteinase K. The resulting solution was centrifuged to remove any undisrupted tissue. The supernatant was then centrifuged, and the pellet resuspended in a specific buffer and incubated on ice for myofibrillar repolymerization. Subsequently, it was centrifuged, the supernatant collected and further centrifuged to generate a mitochondrial pellet. This latter was resuspended in isolation medium. All the steps were performed on ice or at 4 °C. Mitochondrial protein content was then determined by bicinchoninic acid (BCA) assay. 7 µg of mitochondria were loaded in a 24 well Agilent Seahorse XF Cell Culture Microplate and the oxygen consumption rate (OCR) was recorded at the basal level and after the sequential injections of 0.7 mM ADP, 2.25 µg/mL Oligomycin (ATP synthase inhibitor), 10 µM FCCP (uncoupling agent) and a mixture of 3.6 µM Rotenone (complex I inhibitor) and 2 µM Antimycin A (complex III inhibitor).

### 4.6. MRNA Extraction and QPCR Analysis

Total mRNA was extracted by a commercial kit (Qiagen). The cDNA was obtained by reverse-transcription (Maxima First Strand cDNA synthesis kit; Thermo Fisher, Milan, Italy). The qPCRs were performed with Thermo Maxima SYBR Green (Thermo Fisher). Sequences of Tfam and Rpl13a are reported in Table 1.

The following cycling conditions: 95 °C, 10 min; 95 °C, 30 s and 58 °C, 1 min for 40 cycles. The mRNA levels of the genes were expressed with the relative quantity method as 2^−∆∆Ct^.

### 4.7. Western Blot

Skeletal muscle samples harvested after 12 weeks of treatment were homogenized using TissueLyser (Qiagen) in ice-cold lysis buffer with protease and phosphatase inhibitors (10 mM Na orthovanadate, 2 mM phenylmethylsulfonyl fluoride, 20 mM leupeptin, 2 mM benzamidine, 1.5 mM aprotinin) and spun at 14,000× *g* for 10 min at 4 °C. Protein content was measured using a BCA assay. Skeletal muscle extracts containing 30 μg protein were separated by SDS-PAGE (4–12% Bis-Tris gel, NuPAGE). Proteins were transferred to nitrocellulose membranes and were blocked with milk (5%) prior to overnight incubation at 4 °C with primary antibody, as indicated in Table 2. After washing in TBS/T, membranes were incubated for 3 h at room temperature with HRP-conjugated secondary antibodies. Proteins were detected using enhanced chemiluminescence (Biorad). Band intensities were quantified by using Biorad Image J.

### 4.8. Grip and Locomotion Tests

Grip test was performed by using the Grip Meter, which consists of a precision dynamometer connected to a grid, that allows measurement of the strength of limb grasping. Mice were handled by the tail, thus allowing to grasp the grid specifically with their forelimb and pulled out until the grip was left. The grip meter records the maximum weight that the mouse can pull; each mouse was tested for 5 series of pulls composed by 3 consecutives pulls and 1 min of resting in the cage. The average weight of the maximal performance of each series for each mouse was calculated. Strength (gram-force, gf) data was normalized on body weight (g), assessed immediately before the test was conducted. Walking analysis was performed in accordance with De Medinaceli L. [56], and subsequent modifications for footprint quantification. Although this analysis was indicated mainly for the study of nerve damage and regeneration efficacy in peripheral nerve injuries, this method has been applied to several experimental conditions mimicking autoimmune diseases (such as arthritic disorders [57]), stroke [58] and inflammatory neuropathies [59]. The plantar surface of both hind paws was painted with ink and the animal was allowed to walk along a narrow corridor (100 cm long) with white paper on its base. Paper strips were scanned, and, among all parameters, we considered SL, and SW. We measured these parameters, averaging 8–10 footprints per mouse (5 measurements for SL and SW each). Analyses were performed using image J software. Rotarod analysis was performed to test sensory-motor functions and central coordination. Mice were placed on the rod of a rotarod apparatus, which turned on and progressively accelerated (0–40 rpm in 300 s), until the animals fell off. The average time spent on the turning rod, in a set of 3 repetitions, was measured. If the mouse completed the entire test (300 s), the test was not repeated, and maximum time was considered for analysis.

### 4.9. Compound Action Potential (CAP) Recording

Peripheral mice nerves were explanted and CAP was recorded as previously described [60]. After sacrifice, sural and peroneal nerves were removed rapidly by dissection and the epineurium gently removed. Desheathed nerves were mounted between two glass electrodes, in a bath with each nerve stump drawn into a glass pipette and sealed with vaseline, to establish a high resistance electrical seal. The nerve was perfused continuously, with physiological solution (118 NaCl, 3.5 KCl, 20 Hepes, 1.5 CaCl2, 1 MgCl2 and 10 d-glucose, adjusted to pH 7.4) bubbled with 100% O_2_. Pairs of chloride silver wires were used to stimulate the nerve and record extracellular signals. Recordings were performed using the electrophysiology all-in-one rig (Avere solution, Erlangen Germany), signals were amplified, digitized (National Instrument 6341) and processed on-line using QTRAC software (Prof. Hugh Bostock, Digitimer, Hertfordshire, UK). The electrical protocols applied were fully characterized by Bostock and colleagues [61]. The recording of nerve excitability was obtained by monitoring the changes in threshold current required to obtain 40% of maximal compound action potential, during various stimulation protocols: (1) changes of stimulus intensity (stimulus-response); (2) stimulus duration (strength-duration); (3,4) during long-lasting sub-threshold polarizing current pulses (threshold electrotonus and (Threshold I/V); (5) following a single supramaximal stimulus (recovery cycle).

### 4.10. Statistical Analyses

Results are presented as mean ± standard error of the mean (SEM) when normally distributed and as median (interquartile range) when skewed distributed. A *p*-value < 0.05 was considered significant. Group differences were evaluated by Student’s *t*-test (Gaussian distribution) or Mann-Whitney test (skewed distribution). For multiple comparisons, a one-way ANOVA with a 95% confidence interval was used. Grip and locomotion analysis paired outcomes were carried out by repeated measure anova (RM Anova) and analysed the fixed effects variation of time, diet and diet × time. In addition, we performed post-hoc tests matching across time or diet, applying Bonferroni’s multiple comparisons test. Electrophysiological traces originated from 5 electrical stimulation protocols were evaluated by using different approaches: stimulus- response protocol and strength duration protocol outcomes has been subjected to post-analysis. The data obtained (i.e., stimulus-response slope and strength duration time constant) were compared by unpaired *t*-test. On the other hand, the remaining electrical protocol (threshold electrotonus, threshold current/voltage relationship, and recovery cycle) have been analysed comparing alteration in threshold (%) in all the condition tested (stimulus delay, current intensity, and between-stimulus interval). Consequently, 2-way ANOVA has been applied to study differences in nerve excitability considering the two sources of variation of “protocol factor” (stimulus delay, current intensity, and between-stimulus interval) and treatment applied to the experimental group (i.e., atorvastatin) as two independent variables. Bonferroni’s multiple comparisons *post-hoc* test was applied to all 2-way ANOVA analyses.

## 5. Limitations and Conclusions

Data of the present manuscript have to be interpreted in the frame of some limitations. First, the use of a wildtype model would have strengthened the results, although *ApoE^-/-^* is one of the most validated models of preclinical atherosclerosis [24]. Second, since we used both oxidative and glycolytic fibres, the mitochondrial respiration would need to be replicated in the single fibres, e.g., soleus, gastrocnemius, or quadriceps. Third, skeletal muscle fibres composition could have been studied through a histological approach, namely, by using an immunofluorescence co-staining protocol that can simultaneously detect type I, IIa and IIb myosin fibres by means of BA-D5, SC-71 and BF-F3 specific antibodies [62]. Fourth, skeletal muscle composition could be determined by microgenomic approach, based on differences in the expression of genes in slow-oxidative (type 1) and fast-glycolytic (type 2B) fibres, through transcriptome analysis at the single fibre level (microgenomics) [63]. In conclusion, although a recent meta-analysis based on the data from >4 million patients highlighted that the prevalence of complete statin intolerance may be 9.1%, and may be often overestimated [64], the understanding of the mechanisms behind these effects can possibly improve adherence to statin therapy in the patients needed-to-be-treated. Indeed, as reported in a retrospective study of >600,000 patients with established atherosclerotic cardiovascular disease, there is an underuse of high-intensity statins which is likely a major contributor to preventable death and disability [65]. Furthermore, our findings could represent a valid ground to evaluate the impact of other lipid lowering drugs on skeletal muscle mitochondrial activity, locomotion and axonal excitability.

## Figures and Tables

**Figure 1 ijms-23-05415-f001:**
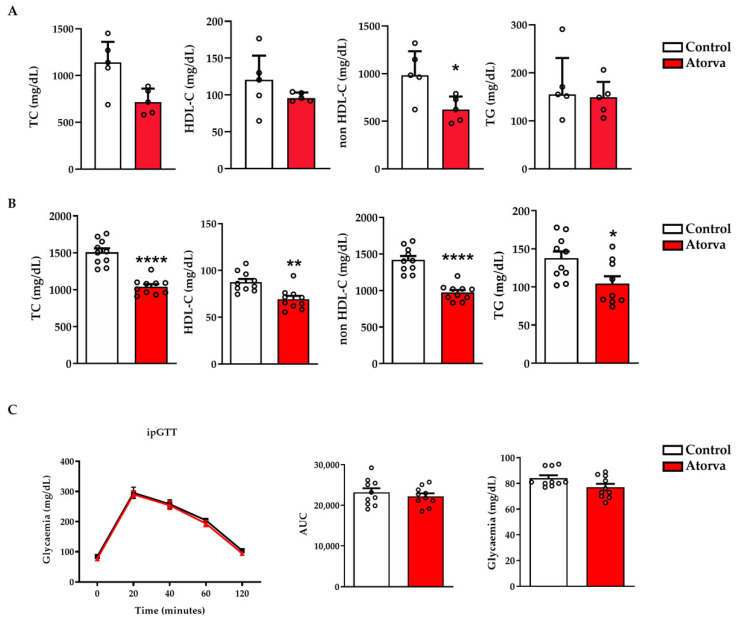
*ApoE^-/-^* mice were fed with HFHC (Control) and HFHC plus atorvastatin (Atorva). Lipid profile was assessed after 1 week (**A**) and 12 weeks (**B**) of diet. Fasting glycaemia after 12 weeks (**C**). Glucose (1 g/kg) was administered intraperitoneally and glycaemia measured after 20, 40, 60 and 120 min. Between-group differences were assessed by Mann–Whitney test (**A**) and by Student’s *t*-test (**B**,**C**). * *p* < 0.05; ** *p* < 0.01; **** *p* < 0.0001. Data in (panel **A**) are depicted as median and interquartile range; those in (**B**,**C**) as mean ± SEM. Atorva, atorvastatin; AUC, area under the curve; HDL-C, high-density lipoprotein cholesterol; HFHC, high-fat high-cholesterol diet; ipGTT, intraperitoneal glucose tolerance test; TC, total cholesterol; TG, triglycerides. *n* = 10 in each group. ° stands for individual values.

**Figure 2 ijms-23-05415-f002:**
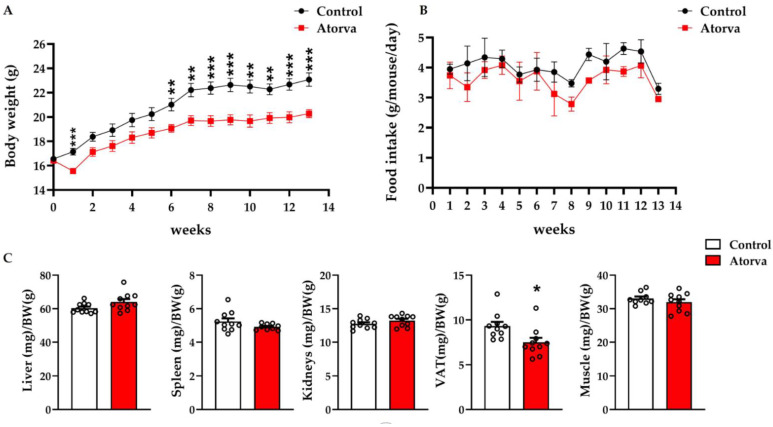
*ApoE^-/-^* mice were fed for 12 weeks with HFHC (Control) and HFHC plus atorvastatin. Body weight (**A**) and food intake (**B**) were recorded weekly. Between-group differences were assessed by Student’s *t*-test. * *p* < 0.05; ** *p* < 0.01; *** *p* < 0.001. Data are depicted as mean ± SEM. Organs were weighed and normalized for body weight (**C**). Atorva, atorvastatin; BW, body weight; HFHC, high-fat high-cholesterol diet; VAT, visceral adipose tissue. *n* = 10 in each group.

**Figure 3 ijms-23-05415-f003:**
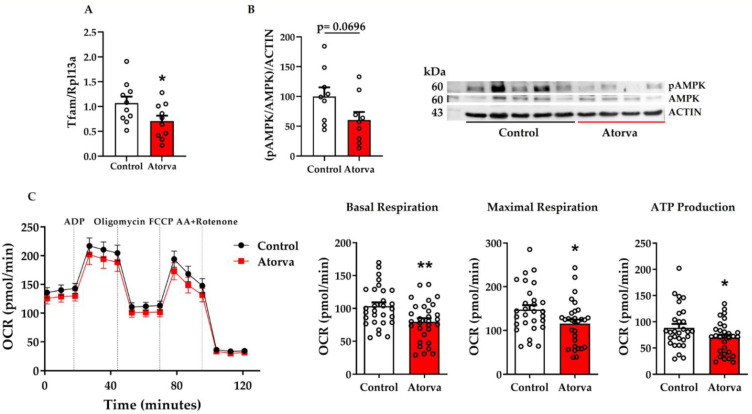
Evaluation of gene expression of *Tfam* (**A**) and representative Western blot pertaining protein expression of pAMPK/AMPK/ACTIN (**B**) in quadriceps. Rpl13a was used as a housekeeping for the gene expression (**A**) and actin was used as a housekeeping for proteins (**B**). Oxygen consumption rate was assessed in mitochondria of skeletal muscles of *ApoE^-/-^* mice fed HFHC and HFHC plus atorvastatin (**C**). Skeletal muscles were a pool of tibialis anterior, extensor digitorum longus, soleus, gastrocnemius, quadriceps, and biceps brachii. Between-group differences were assessed by Student’s *t*-test. * *p* < 0.05; ** *p* < 0.01. Data are expressed as mean ± SEM. ADP, adenosine triphosphate; AA, antimycin A; FCCP, carbonyl cyanide-*p*-trifluoromethoxyphenyl-hydrazone; HFHC, high-fat high-cholesterol diet; OCR, oxygen consumption rate; Tfam, mitochondrial transcription factor; AMPK, serine/threonine kinase AMP-activated protein kinase. *n* = 9 in each group (**A**,**B**); *n* = 6 in each group (**C**). Control = high-fat high-cholesterol diet; Atorva = high-fat high-cholesterol diet plus atorvastatin. kDa, kilodalton; SEM, standard error of the mean.

**Figure 4 ijms-23-05415-f004:**
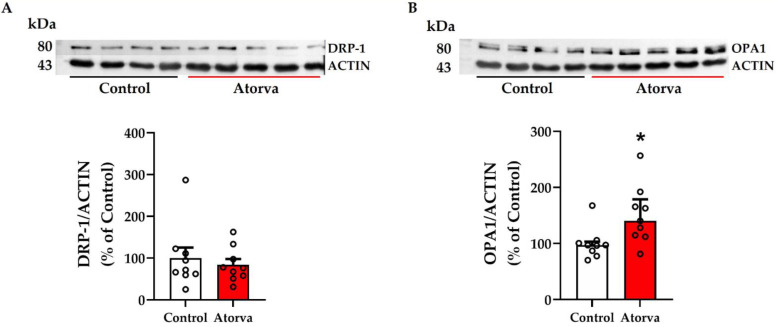
Representative blots pertaining to protein expression of DRP-1 (**A**) and OPA1 (**B**) in quadriceps of *ApoE^-/-^* mice fed HFHC and HFHC plus atorvastatin. Actin was used as a housekeeping. Between-group differences were assessed by Student’s *t*-test (**A**) and Mann-Whitney (**B**). * *p* < 0.05. Data are expressed as mean ± SEM (**A**) and median and interquartile range (**B**). DRP-1, dynamin related protein 1; OPA1, Optic atrophy 1. *n* = 9 in each group. Control = high-fat high-cholesterol diet (HFHC); Atorva = high-fat high-cholesterol diet plus atorvastatin. kDa, kilodalton. SEM, standard error of the mean.

**Figure 5 ijms-23-05415-f005:**
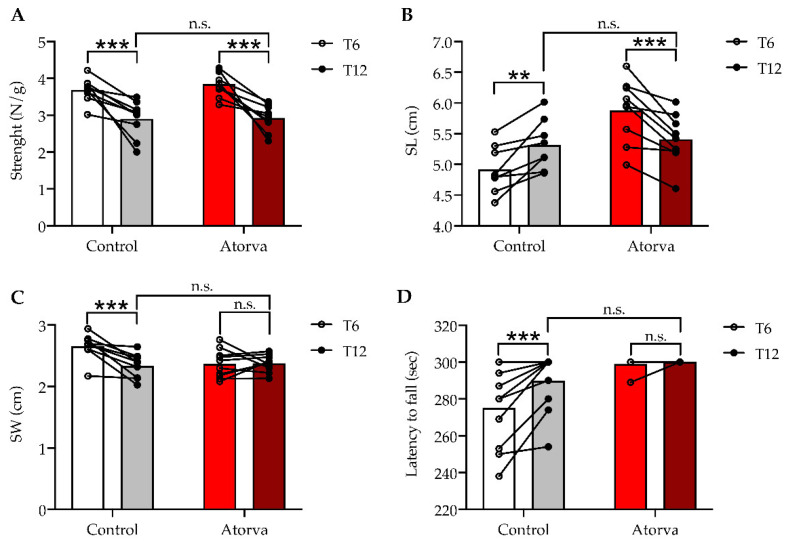
Muscle strength and locomotion. Grip test was used for muscle strength and data were expressed as strength per animal weight (g). RM ANOVA (paired analysis) *n* = 9 pairs, time factor **** *p* < 0.0001 F(1.16) = 47.48; diet factor n.s. *p* = 0.4856 F(1.17) = 0.5081; diet × time factor n.s. *p* = 0.5811 F(1.16) = 0.3172; Bonferroni’s multiple comparison test (control T6 vs. control T12) *** *p* = 0.0009, *t* = 4.419, df = 16, (atorva T6 vs. atorva T12) *** *p* = 0.0001, *t* = 5.338, df = 16; (control T12 vs. atorva T12) n.s. *p* > 0.9999 *t* = 0.1167, df = 33 (**A**). Locomotion was assessed by walking analysis behavioural test. Stride length (SL) and stride width (SW) were obtained averaging 5 consequent steps for each animal; SL differences has been analysed by RM ANOVA (paired analysis) *n* = 8 pairs, time factor n.s. *p* = 0.6156 F(1.15) = 6.997; diet factor * *p* = 0.0184 F(1.15) = 6.997; diet × time factor **** *p* < 0.0001 F(1.15) = 35.6, Bonferroni’s multiple comparison test (control T6 vs. control T12) ** *p* = 0.0039, *t* = 3.748, df = 15, (atorva T6 vs. atorva T12) *** *p* = 0.0005, *t* = 4.723, df = 15; (control T12 vs. atorva T12) n.s. *p* > 0.9999 *t* = 0.4322, df = 30 (**B**). Differences in SW were been analysed applying RM ANOVA (paired analysis) *n* = 9 pairs, time factor ** *p* = 0.0071 F(1.17) = 9.348; diet factor n.s. *p* = 0.1155 F(1.17) = 2.751; diet × time factor ** *p* = 0.005 F(1.17) = 10.36, Bonferroni’s multiple comparison test (control T6 vs. control T12) *** *p* = 0.0009, *t* = 4.325, df = 17, (atorva T6 vs. atorva T12) n.s. *p* > 0.9999, *t* = 0.1167, df = 17; (control T12 vs. atorva T12) n.s. *p* > 0.9999 *t* = 0.4633, df = 34 (**C**). Coordination was assessed by rotarod test. RM ANOVA (paired analysis) *n* = 9 pairs, time factor ** *p* = 0.0017 F(1.18) = 13.58; diet factor ** *p* = 0.008 F(1.18) = 8.889; diet × time factor ** *p* = 0.0053 F(1.18) = 10.06, Bonferroni’s multiple comparison test (control T6 vs. control T12) *** *p* = 0.0003, *t* = 4.848, df = 18, (atorva T6 vs. atorva T12) n.s. *p* > 0.9999, *t* = 0.3628, df = 18; (control T12 vs. atorva T12) n.s. *p* = 0.2054 *t* = 1.674, df = 36. Control = high-fat high-cholesterol diet; Atorva = high-fat high-cholesterol diet plus atorvastatin. Bars in white and bars in red represent T6; bars in grey and cherry red represent T12 (**D**).

**Figure 6 ijms-23-05415-f006:**
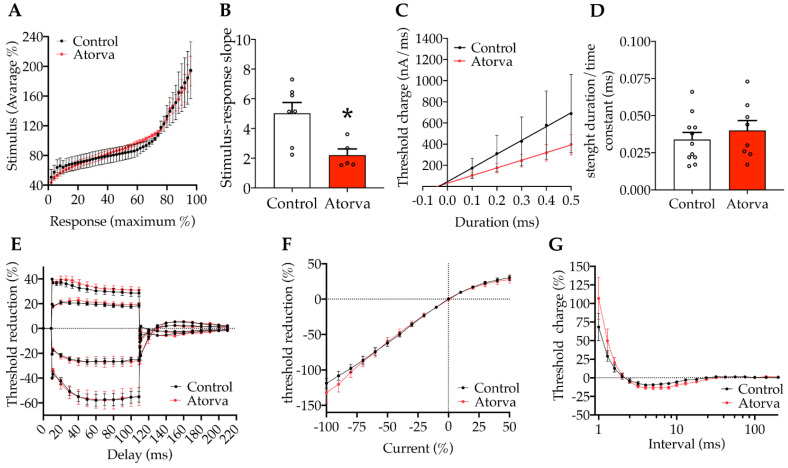
Nerve conduction. Stimulus-response recording shows the relationship between the % of supramaximal compound action potential (CAP) response evoked by crescent current stimuli, normalized for the complex average stimulus applied. Plotted representation of stimulus-response curves (**A**). Stimulus-response outcomes were analysed by comparing sigmoidal’s slope values calculated for each nerve. (*n* = 8, unpaired *t* test * *p* = 0.0128, *t* = 3.023, df = 10) (**B**). Strength-duration protocol assesses the total energy of a stimulus required by the fibre to generate an action potential, measured as the current required to elicit 40% of CAP by different stimulus duration. Plotted representation of strength duration outcomes (**C**); from 0.1 ms to 0.5 ms square pulse. The strength-duration time constant (*τ*_SD_) represents the intercept on the stimulus duration axis. *τ*_SD_ values as been analysed (Control (*n* = 11); Atorva (*n* = 8); unpaired *t*-test n.s. *p* = 0.4593, *t* = 0.7572, df = 17 (**D**). In the threshold electrotonus protocol, percent reduction was calculated as the peak of reduction between those obtained before and after subthreshold depolarizing or hyperpolarizing currents, averaged over 20 ms. Four different protocols were used as represented in panel **E** (±20% and ±40% currents). 2way ANOVA test was applied to each electrical protocol, comparing control (*n* = 10) and atorva (*n* = 7). +40% treatment variation *p* = 0.0509, F (1.420) = 3.833; +20% treatment variation *p* = 0.2338 F(1.390) = 1.422; −20% treatment variation *p* = 0.4477 F(1.390) = 0.5776; −40% *p* = 0.9770, F(1.390) = 0.0008 (**E**). In the threshold I/V protocol current-threshold relationship was calculated by comparing conditioned threshold peak measured after 200 ms depolarizing to hyperpolarizing subthreshold current applied to a resting condition. Control (*n* = 10), atorva (*n* = 7); 2way ANOVA test was applied (*p* = 0.2160) (**F**). Recovery cycle protocol was used to assess refractoriness and super excitability as the percentage change in threshold after supramaximal conditioning stimulus at crescent interstimulus intervals. control (*n* = 8), atorva (*n* = 7); 2way ANOVA test was applied (*p* = 0.4472 F(1.268) = 0.5795) (**G**). Control = high-fat high-cholesterol diet; Atorva = high-fat high-cholesterol diet plus atorvastatin.

**Figure 7 ijms-23-05415-f007:**
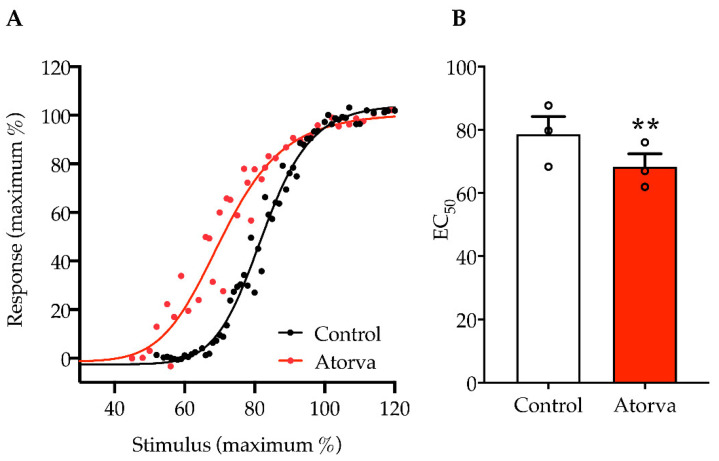
Stimulus-response protocol recordings of C-CAP. Stimulus intensity applied, and C-CAP response measured is presented as the relationship between the fold supramaximal CAP response evoked by crescent current stimuli, normalized on the maximal stimulus applied (expressed in %). Plotted representation of stimulus-response curves (**A**). Stimulus-response curve EC_50_ variation has been statistically analysed. Control (*n* = 3); Atorva (*n* = 3); unpaired *t*-test, ** *p* = 0.0031; *t* = 6.4; df = 4 (**B**). Control = high-fat high-cholesterol diet; Atorva = high-fat high-cholesterol diet plus atorvastatin.

**Figure 8 ijms-23-05415-f008:**
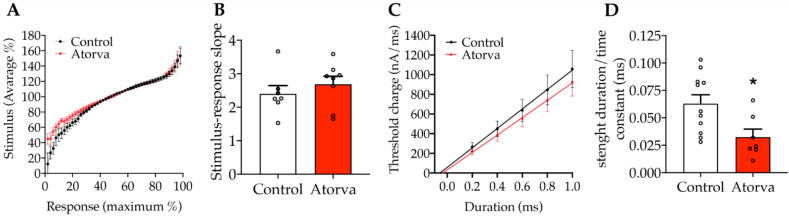
Stimulus-response protocol recordings are presented as the relationship between the % of supramaximal CAP response evoked by crescent current stimuli, normalized on the complex average stimulus applied. Plotted representation of stimulus-response curves (**A**). This parameter was statistically analysed comparing stimulus-response slope values (**B**); Control *n* = 7, Atorva *n* = 8; unpaired *t*-test n.s. *p* = 0.4205, *t* = 0.8318, df = 13). Threshold modulation mediated by different charge duration (from 0.2 ms to 1.0 ms square pulse) is presented as raw threshold measurement (**C**). The comparison between experimental conditions was performed on time constant values (**D**); control *n* = 11, atorva *n* = 7; unpaired *t*-test * *p* = 0.0189, *t* = 2.610, df = 16. *n* = 9. Control = high-fat high-cholesterol diet; Atorva = high-fat high-cholesterol diet plus atorvastatin.

**Figure 9 ijms-23-05415-f009:**
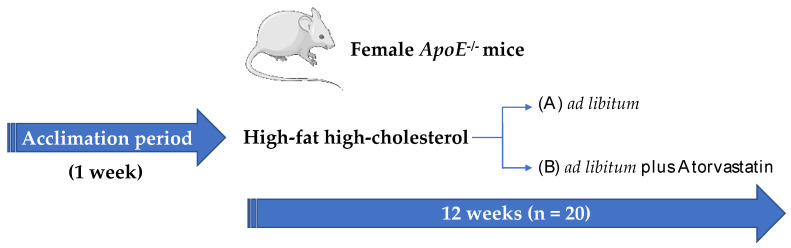
All animal procedures performed conform to the guidelines from directive 2010/63/EU of the European Parliament on the protection of animals and were approved by the Ethical Committee (authorization number 476/2020-PR). At sacrifice, liver, visceral adipose tissues, skeletal muscles, spleen, and kidneys were collected and weighted.

**Table 1 ijms-23-05415-t001:** Primers used for gene expression analysis.

Gene	FORWARD (5′→3′)	REVERSE (5′→3′)
Rpl13a	GCGCCTCAAGTGGTGTTGGAT	GAGCAGCAGGGACCACCAT
Tfam	CGGGCCATCATTCGTCG	AGACAAGACTGATAGACGAGGG

**Table 2 ijms-23-05415-t002:** Antibodies characteristics.

Primary Antibody (Dilution)	Secondary Antibody (Dilution)
pAMPK (1:1000)	anti-rabbit (1:10,000)
AMPK (1:1000)	anti-mouse (1:10,000)
DRP-1 (1:1000)	anti-mouse (1:10,000)
OPA1 (1:1000)	anti-rabbit (1:10,000)
Actin (1:1000)	anti-mouse (1:10,000)

AMPK, serine/threonine kinase AMP-activated protein kinase; pAMPK, phosphorylated AMPK; DRP-1, dynamin related protein 1; OPA1, optic atrophy 1.

## Data Availability

Data are available upon request.

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
