# Peer review of "Impact of Atorvastatin on Skeletal Muscle Mitochondrial Activity, Locomotion and Axonal Excitability—Evidence from ApoE-/- Mice"

_ijms, 2022, doi:10.3390/ijms23105415_

Round 1

Reviewer 1 Report

Overall I found this to be a very interesting and well down study. I particularly appreciate the molecular, cellular, and functional measurements. Very thorough, very comprehensive and important work.  I have a few comments and suggestions.

Minor

  1. I don't really come away from the introduction understanding why you are looking at the neural pathways. I think it would help the reader if you made that connection more obvious.
  2. The use of abbreviations takes away from the overall readability of your paper. I read the paper over many time and I still don't recall what SAMS, ASCVD and RST are. If I recall correctly you only used SAMS more than once ( I do realize the abbreviation SAMS is common) and the others only once. In short you paper would read better if you eliminated some of the abbreviations. 
  3. The figure are jammed with data, which on some level is good because you paper is comprehensive but I struggled with the take away points. 

Potentially more serious.

  1. Why group the analysis of all muscles? Is it possible that the impact is fiber type specific? I ask because the variability of the responses in you treatment group compared to the control. In particular figure #3 the difference in the structure of the data is striking. Your Control group is heavily leftward skewed while your Treatment group is highly variable. What do you think the reason for this is? 
  2. Related to that, and I do not know the answer to this off the top of my head but, are your statistical tests robust to distribution like these?

Author Response

Overall I found this to be a very interesting and well down study. I particularly appreciate the molecular, cellular, and functional measurements. Very thorough, very comprehensive and important work.  I have a few comments and suggestions.

We thank the reviewer for this thoughtful comment.

Minor

  1. I don't really come away from the introduction understanding why you are looking at the neural pathways. I think it would help the reader if you made that connection more obvious.

We thank the reviewer for this comment that was well taken. The part relative to neuropathy has been improved and now reads as follows “Another aspect worthy of consideration is the impact of statin use on the rate of peripheral neuropathy, a common clinical problem affecting at least 8% of the population >55 years of age [15], even greater if considering only peripheral nerve injuries. This condition, characterized by dysfunction and damage to peripheral nerves, appear to be dependent on numerous factors and occurs typically with lipophilic statins. Peripheral neuropathy may affect different fibres due to different aetiologies and can be classified according to the type of fibre affected and the pattern of injury [16], as in the Guillain-Barrè syndrome found upon initiation of statin therapy and characteristically associated with altered nerve conduction [17]. Although observational studies suggest a possible association between statin use (prior or current) and newly diagnosed peripheral neuropathy [18,19], primary peripheral neuropathy is substantially more common and according to some, even more frequent than muscle injury [18], leading to one excess case of idiopathic peripheral neuropathy for every 2,200 (880 to 7,300) person-years of statin use [20]. However, caution has been expressed on the validity of these findings [21]. Indeed, there is no support either for a causal relationship in randomized controlled trials [22] or from an aetiological point of view. Neuronal cholesterol biosynthesis impairment does in fact occur with a lipophilic statin such as simvastatin, with reduced synapse density and impaired synaptic vesicle release even when lipoproteins or geranylgeraniol are added to the system [23]. It appears therefore of interest to evaluate neuropathy in standardized conditions in an appropriate animal model.”

  1. The use of abbreviations takes away from the overall readability of your paper. I read the paper over many time and I still don't recall what SAMS, ASCVD and RST are. If I recall correctly you only used SAMS more than once ( I do realize the abbreviation SAMS is common) and the others only once. In short you paper would read better if you eliminated some of the abbreviations.

We thank the reviewer for this advice. The text has been revised accordingly.

  1. The figure are jammed with data, which on some level is good because you paper is comprehensive but I struggled with the take away points.

We have tried to make figures self-explanatory as much as possible.

Potentially more serious.

  1. Why group the analysis of all muscles? Is it possible that the impact is fiber type specific?

 I ask because the variability of the responses in you treatment group compared to the control. In particular figure #3 the difference in the structure of the data is striking. Your Control group is heavily leftward skewed while your Treatment group is highly variable. What do you think the reason for this is? 

The variability of the distribution is an intrinsic drawback of using animal models and this issue was accentuated by extracting alive mitochondria from fibres. We were not able to extract a sufficient amount of mitochondria from the single fibres, thus, based on previous published manuscripts by the first Authors (JCI Insight. 2019 Mar 7;4(5):e123130. doi: 10.1172/jci.insight.123130; Sci Rep. 2021 Feb 26;11(1):4789. doi: 10.1038/s41598-021-84213-w), we have chosen the approach to extract mitochondria from a pool of tibialis anterior, extensor digitorum longus, soleus, gastrocnemius, quadriceps, and biceps brachii.  All the data have been normalized for the total quantity of proteins loaded. Since we agree with the reviewer that we used both oxidative and glycolytic fibres, this limitation has been listed among others in a dedicated section that reads “Data of the present manuscript have to be interpreted in the frame of some limitations. First, the use of a wildtype model would have strengthened the results, although ApoE-/- is one of the most validated models of preclinical atherosclerosis [24]. Second, since we used both oxidative and glycolytic fibres, the mitochondrial respiration would need to be replicated in the single fibres, e.g., soleus, gastrocnemius, or quadriceps. Third, skeletal muscle fibres composition could have been studied through a histological approach, namely, by using an immunofluorescence co-staining protocol that can simultaneously detect type I, IIa and IIb myosin fibres by means of BA-D5, SC-71 and BF-F3 specific antibodies [62]. Fourth, skeletal muscle composition could be determined by microgenomic approach, based on differences in the expression of genes in slow-oxidative (type 1) and fast-glycolytic (type 2B) fibres, through transcriptome analysis at the single fibre level (microgenomics) [63].”

  1. Related to that, and I do not know the answer to this off the top of my head but, are your statistical tests robust to distribution like these?

We addressed this question of yours in the Statistical section of the paper, concluding that, in view of the data distribution (skewed or Gaussian), appropriate tests have been used, namely, Mann-Whitney or Student’s t-test.

Reviewer 2 Report

The authors report physiological assays to assess the effect of atorvastatin treatment on muscle function in Apoe-/- mouse model.

It is a good and relevant study.

When discussing muscle physiology, myofiber-typing is a robust readout to assess changes in contraction. I find it relevant to add this data, if not possible, please add to the discussion.

My major critics concerns the statistical tests that were employed to assess an effect. The inconsistencies is worrying, specific comments are listed in the comments to the results’ section.  

Abstract

Please explain in the background why using the Apoe-/- mouse model to investigate the posted question.

Write the methods as sentences, instead of a list. Indicate the age of the mice, number of mice, time range of the treatment.

The time range should be indicated in the results. This could also be relevant to the conclusions.

Introduction:

Clearly written, yet – the choice to employ the Apoe-/- mouse model to address the specific question is not sufficiently described.

Results:

  • Line 73: “to validate” – unclear. Validation is of previous results. Are the presented results in Fig. 1 are novel or not?.
  • 1: indicate the nr of mice in each test. Explain the choice to test an effect after 12 weeks? I suggest incorporating Fig. S1 in the main text (Fig. 1A or B?). This shows a progressed effect between 1 and 12 weeks.
  • It is unclear why the authors choose for the Mann–Whitney test, which has the least power?
  • Line 106: gene/protein name should be for mouse (Tfam) not for human (TFAM).
  • The WB is Fig. 2B and suppl Fig 2 (pAmpk) are inconsistent, in suppl Fig. 2 there is no change. Please show a third blot. The choice for the Mann–Whitney test to assess if the difference between the groups in Fig. 3 is unclear. Specifically, the Mann–Whitney test is mostly applied for a small nr of samples with similar variations, which is not the case in 2C.
  • Line 129 should be Suppl Fig. 2. The WB is Fig 4 shows different results from the WB in Suppl Fig. 2 (same comment as above).
  • Line 149 should be Suppl Fig. 3.
  • The presentation of the results in Fig. 5 is confusing and unclear: in brief, the two-group comparison can be combined in a time-dependent analysis, and multiple testing (anova) can assess statistical difference between all conditions. The conclusions will be clearer. What A, D, G, J represent is it T0?
  • The choice for anova test is Fig 6 is logical, but why it is the only experiment where this test was applied to assess a statistical difference? In Fig. 7 and Fig. 8 it is the unpaired t-test. The authors should be consistent or explain the choice for each test.

Discussion:

The discussion should be adjusted after applying a correct statistical test. The discussion is highly speculative, and statements should be tuned down.

The paper mentioned in the conclusion [57] should be discussed in more details in the discussion, and how the results in the presented study support or not support the conclusion in the paper [57].

Author Response

Reviewer#2

Abstract

  1. Please explain in the background why using the Apoe-/- mouse model to investigate the posted question. Write the methods as sentences, instead of a list. Indicate the age of the mice, number of mice, time range of the treatment. The time range should be indicated in the results. This could also be relevant to the conclusions.

We thank the reviewer for this advice, it was well taken and we took care of these issues.

Introduction:

  1. Clearly written, yet – the choice to employ the Apoe-/- mouse model to address the specific question is not sufficiently described.

We thank the reviewer for this advice. References 24 and 25 (Emini Veseli, B et al., Eur J Pharmacol 2017, 816, 3-13 and Bjorkegren, J.; Lusis, A.J. Atherosclerosis: Recent developments. Cell 2022) were added to justify the model. Indeed, on a high-cholesterol diet, ApoE-/- mice develop plaques more rapidly and with a more advanced phenotype as compared to LDLr-/- mice

Results:

  1. Line 73: “to validate” – unclear. Validation is of previous results. Are the presented results in Fig. 1 are novel or not?.

We thank the reviewer for pointing this out. Due to the variation among animal housing in terms of behavioural environment stresses, we wanted to evaluate the effectiveness of the experimental paradigm in our hands.

  1. 1: indicate the nr of mice in each test.

This info is now reported in each figure legend

  1. Explain the choice to test an effect after 12 weeks?

This time frame was chosen according to a study regarding the potentiality of bempedoic acid to avoid the myotoxicity associated with statin therapy (Pinkosky SL et al., Nat Commun. 2016 Nov 28;7:13457). New Ref. 55.

  1. I suggest incorporating Fig. S1 in the main text (Fig. 1A or B?). This shows a progressed effect between 1 and 12 weeks.

We thank the reviewer for this suggestion and now the formerly Figure S1 has been moved to Fig. 1 as panel A.

  1. It is unclear why the authors choose for the Mann–Whitney test, which has the least power?

We addressed this question of yours in the Statistical section of the paper, concluding that, in view of the data distribution (skewed or Gaussian), appropriate tests have been used, namely, Mann-Whitney or Student’s t-test.

  1. Line 106: gene/protein name should be for mouse (Tfam) not for human (TFAM).

We thank the reviewer. The text has been amended accordingly.

  1. The WB is Fig. 2B and suppl Fig 2 (pAmpk) are inconsistent, in suppl Fig. 2 there is no change. Please show a third blot. The choice for the Mann–Whitney test to assess if the difference between the groups in Fig. 3 is unclear. Specifically, the Mann–Whitney test is mostly applied for a small nr of samples with similar variations, which is not the case in 2C.

Apologies for not making this concept clear. The graph bars of Figure 3B, Figure 4A-B report nine dots which refer to the analysis of mitochondrial proteins extracted from quadriceps of 9 mice fed HFHC vs 9 mice fed HFHC plus atorvastatin. In the former supplementary Figure 2, now supplementary Figure 1, we reported the Western blot analyses of all samples. However, the statistical analysis was performed by using the densitometry analyses of all nine samples. In addition, data on pAMPK have been also normalized for the non-phosphorylated form, namely, AMPK. The text reads “First, we assessed the impact of atorvastatin on mitobiogenesis in quadriceps. In mice given atorvastatin, gene expression of mitochondrial transcription factor (Tfam) was significantly reduced (Figure 3A), whereas the phosphorylated protein levels of serine/threonine kinase AMP-activated protein kinase (AMPK) showed a trend toward reduction (p= 0.0696; Figure 3B). The protein expression of all replicates is reported in Supplementary Fig. 1A.”

  1. Line 129 should be Suppl Fig. 2. The WB is Fig 4 shows different results from the WB in Suppl Fig. 2 (same comment as above).

We hope we have clarified enough the issue in the previous answer.

  1. Line 149 should be Suppl Fig. 3.

  Apologies for this mistake. The text has been amended accordingly.

  1. The presentation of the results in Fig. 5 is confusing and unclear: in brief, the two-group comparison can be combined in a time-dependent analysis, and multiple testing (anova) can assess statistical difference between all conditions. The conclusions will be clearer. What A, D, G, J represent is it T0?

We thank the reviewer for this suggestion that was well taken. Accordingly, Figure 5 now combines the split graphs in grouped ones. We performed multiple comparison statistical tests. Given the time-dependent nature of the follow-up of behavioural outcomes of mice in the two experimental groups, we kept paired representation and statistical analysis. Therefore, we performed the repeated measure ANOVA (RM ANOVA) and analysed the fixed effects variation of time, diet and diet x time. In addition, we performed post-hoc tests matching across time or diet, applying Bonferroni’s multiple comparisons test. We confirmed all the reported observations and adjusted result section and Fig 5 legend accordingly.

Figure 5: Muscle strength and locomotion. Grip test was used for muscle strength and data were expressed as strength per animal weight (g). RM ANOVA (paired analysis) n=9 pairs, time factor **** p<0.0001 F(1,16)= 47.48; diet factor n.s. p=0.4856 F(1,17)= 0.5081; diet x time factor n.s. p=0.5811 F(1,16)= 0.3172; Bonferroni’s multiple comparison test (control T6 vs. control T12) *** p=0.0009, t= 4.419, df= 16, (atorva T6 vs. atorva T12) *** p=0.0001, t=5.338, df=16; (control T12 vs. atorva T12) n.s. p>0.9999 t=0.1167, df=33 (A). Locomotion was assessed by walking analysis behavioural test and stride length (SL) and stride width (SW) were obtained averaging 5 consequent steps for each animal; SL differences has been analysed by RM ANOVA (paired analysis) n=8 pairs, time factor n.s. p=0.6156 F(1,15)= 6.997; diet factor * p=0.0184 F(1,15)= 6.997; diet x time factor **** p<0.0001 F(1,15)=35.6, Bonferroni’s multiple comparison test (control T6 vs. con-trol T12) ** p=0.0039, t= 3.748, df= 15, (atorva T6 vs. atorva T12) *** p=0.0005, t=4.723, df=15; (control T12 vs. atorva T12) n.s. p>0.9999 t=0.4322, df=30 (B). Differences in SW has been ana-lysed applying RM ANOVA (paired analysis) n=9 pairs, time factor ** p=0.0071 F(1,17)=9.348; diet factor n.s. p=0.1155 F(1,17)=2.751; diet x time factor ** p=0.005 F(1,17)=10.36, Bonferroni’s multiple comparison test (control T6 vs. control T12) *** p=0.0009, t= 4.325, df= 17, (atorva T6 vs. atorva T12) n.s. p>0.9999, t=0.1167, df=17; (control T12 vs. atorva T12) n.s. p>0.9999 t=0.4633, df=34 (C). Coordination was assessed by rotarod test. RM ANOVA (paired analysis) n=9 pairs, time factor ** p=0.0017 F(1,18)=13.58; diet factor ** p=0.008 F(1,18)=8.889; diet x time factor ** p=0.0053 F(1,18)=10.06, Bonferroni’s multiple comparison test (control T6 vs. control T12) *** p=0.0003, t= 4.848, df= 18, (atorva T6 vs. atorva T12) n.s. p>0.9999, t=0.3628, df=18; (control T12 vs. atorva T12) n.s. p=0.2054 t=1.674, df=36. Atorva, high-fat high cholesterol diet plus atorvastatin; Control, high-fat high cholesterol diet.

  1. The choice for anova test is Fig 6 is logical, but why it is the only experiment where this test was applied to assess a statistical difference? In Fig. 7 and Fig. 8 it is the unpaired t- The authors should be consistent or explain the choice for each test.

Apologies for not making this concept clear and we thank the reviewer for the opportunity to clarify the statistical approach applied in electrophysiological experiments.

Electrophysiological protocols have been analysed by applying different statistical approaches depending on specific functional outcomes. Stimulus-response protocol, qualitatively represented in Fig 6A, has been statistically analysed in terms of curve slope values (Fig 6B) and unpaired t-test has been applied. Similarly, charge duration protocol, qualitatively represented in Fig 6C, generates different values of strength duration time constant that have been represented in Fig 6D and statistically analysed as single values for preparation, thus applying unpaired t-test. On the other hand, electrical protocol, such as threshold electrotonus (fig. 6E), threshold current/voltage relationship (Fig 6F), and recovery cycle (Fig 6G) are characterized by the study of the entire protocol (and not only by an extrapolated factors), analysing the alteration in threshold (%) in all the condition tested (stimulus delay, current intensity, and between-stimulus interval). Consequently, 2way ANOVA has been applied to study differences in nerve excitability considering the tow sources of variation of “protocol factor” (again depending on stimulus delay, current intensity, and between-stimulus interval) and “diet/treatment factor”.

In figure 7 and figure 8, we reported only the stimulus-response protocol and charge duration protocol (Overlapping figure 6A, 6B, 6C and 6D) applied on C-fibres and Aa fibres, respectively. We consistently applied unpaired t-test to study the effect of the diet on the factor/constants originated by these protocols (Figure 7B and 8B, 8D).

We integrated the figure legends with more details to clarify these differences.

Discussion:

The discussion should be adjusted after applying a correct statistical test. The discussion is highly speculative, and statements should be tuned down.

We tried to deal with this issue.

The paper mentioned in the conclusion [57] should be discussed in more details in the discussion, and how the results in the presented study support or not support the conclusion in the paper [57].

We thank the reviewer for this comment. Based on our data, we cannot and we do not want to confirm or deny the meta-analysis by the Group of Prof. Banach. Our objective was to stress that it is important to understand in detail mechanisms behind statin-driven SAMS also considering that another lipid lowering drug, i.e., bempedoic acid has been proposed not to lead to SAMS since it affects the pathway of cholesterol biosynthesis only in the liver and not in skeletal muscles.

The conclusions now read “In conclusion, although a recent meta-analysis based on the data from > 4 million patients highlighted that the prevalence of complete statin intolerance may be 9.1%, and may be often overestimated [64], the understanding of the mechanisms behind these effects can possibly improve adherence to statin therapy in the patients needed-to-be-treated. In-deed, as reported in a retrospective study of > 600,000 patients with established ASCVD, there is an underuse of high-intensity statins which is likely a major contributor to preventable death and disability [65]. Furthermore, our findings could represent a valid ground to evaluate the impact of other lipid lowering drugs on skeletal muscle mitochondrial activity, locomotion and axonal excitability.”

Reviewer 3 Report

In the article “impact of atorvastatin on skeletal muscle mitochondrial activity, locomotion and axonal exitability – evidence from Apoe-/- mice” Macchi and colleagues showed that in Apoe-/- mice fed with high fat high cholesterol diet (model of hypercholesterolemic mice) the treatment with Atorvastatin did not affect muscular strength, reduced muscle mitochondrial biogenesis and altered the responsiveness of mechanoceptive and nociceptive fibres.

Before the publication in International Journal of Molecular Sciences, the authors should address some modifications.

Minor points:

  • Line 108 and 130: the figure involved is the Supplementary 2;
  • Line 149: the figure involved is the Supplementary 3;
  • In the figure legend of the Supplementary 1 “**p<0.01; ***p<0.001; ****p<0.0001” are not necessary;
  • In line 148: please change “stride and width length” with stride width and length as is written in the figure legend;
  • In line 336 there is a double full stop at the end of a sentence.

Major points:

  • First of all I suggest to include a group of aged-matched wild type mice, especially for the mitochondrial respiration analysis, for the protein expression and the muscle performance tests;
  • Please specify in which tissues are analyzed Tfam gene expression and pAmpk protein expression;
  • In which skeletal muscles of Apoe-/- mice did you test DRP-1 and OPA1 of Figure 4?;
  • Is it possible to show separately the skeletal muscle weight? e. tibialis anterior, gastrocnemius exc.;
  • Since the authors observed that atorvastatin influenced the fibers responsiveness might be interesting to analyze if this aspect should induce some modifications in the skeletal muscle fiber composition, thus an experiment to check the slow and fast myofibers should be added.

Author Response

Reviewer#3

In the article “impact of atorvastatin on skeletal muscle mitochondrial activity, locomotion and axonal exitability – evidence from Apoe-/- mice” Macchi and colleagues showed that in Apoe-/- mice fed with high fat high cholesterol diet (model of hypercholesterolemic mice) the treatment with Atorvastatin did not affect muscular strength, reduced muscle mitochondrial biogenesis and altered the responsiveness of mechanoceptive and nociceptive fibres.

Before the publication in International Journal of Molecular Sciences, the authors should address some modifications.

Minor points:

  • Line 108 and 130: the figure involved is the Supplementary 2;

Apologies for this mistake. However, since Supplemental Figure 1 has been moved to Figure 1, the old Supplemental Figure 2 becomes Supplemental Figure 1.

  • Line 149: the figure involved is the Supplementary 3;

Apologies, the text has been amended accordindly to the previous answer and now the former Supplementary Figure 3 is Supplementary Figure 2.

  • In the figure legend of the Supplementary 1 “**p<0.01; ***p<0.001; ****p<0.0001” are not necessary;

Thank you. Supplemental Figure 1 has been now moved to Figure 1.

  • In line 148: please change “stride and width length” with stride width and length as is written in the figure legend;

The text has been amended accordingly and reads “stride width and stride length”

  • In line 336 there is a double full stop at the end of a sentence.

Thank you. The text has been amended accordingly.

Major points:

  • First of all I suggest to include a group of aged-matched wild type mice, especially for the mitochondrial respiration analysis, for the protein expression and the muscle performance tests;

As you suggested, a wildtype group would have strengthened the conclusions. Unfortunately, this is not feasible at this stage as the animal licence for this project requires renewal and it would take us few months before being able to perform such experiments.

We have highlighted this aspect within the limitations section.

  1. Limitations and Conclusions. Data of the present manuscript have to be interpreted in the frame of some limitations. First, the use of a wildtype model would have strengthened the results, although ApoE-/- is one of the most validated models of preclinical atherosclerosis [24]. Second, since we used both oxidative and glycolytic fibres, the mitochondrial respiration would need to be replicated in the single fibres, e.g., soleus, gastrocnemius, or quadriceps. Third, skeletal muscle fibres composition could have been studied through a histological approach, namely, by using an immunofluorescence co-staining protocol that can simultaneously detect type I, IIa and IIb myosin fibres by means of BA-D5, SC-71 and BF-F3 specific antibodies [62]. Fourth, skeletal muscle composition could be determined by microgenomic approach, based on differences in the expression of genes in slow-oxidative (type 1) and fast-glycolytic (type 2B) fibres, through transcriptome analysis at the single fibre level (microgenomics) [63]. In conclusion, although a recent meta-analysis based on the data from > 4 million patients highlighted that the prevalence of complete statin intolerance may be 9.1%, and may be often overestimated [64], the understanding of the mechanisms behind these effects can possibly im-prove adherence to statin therapy in the patients needed-to-be-treated. Indeed, as re-ported in a retrospective study of > 600,000 patients with established ASCVD, there is an underuse of high-intensity statins which is likely a major contributor to preventable death and disability [65]. Furthermore, our findings could represent a valid ground to evaluate the impact of other lipid lowering drugs on skeletal muscle mitochondrial activity, locomotion and axonal excitability.

However, for your convenience, we report an analysis of mitochondrial functionality of wildtype mice fed chow diet pertaining another project we are running. The mitochondrial respiration below depicted is superimposable to the one reported in the present manuscript, as shown by the shape of the curve. Responses to drugs are similar, namely, there is a rise in mitochondrial respiration upon injections of ADP and FCCP and a decrement upon injection of oligomycin and rotenone.

  • Please specify in which tissues are analyzed Tfam gene expression and pAmpk protein expression;

Quadriceps. Apologies for not reporting this info in the original version. It has now been added into the text and figure legend of Figure 3A-B and Figure 4A-B.

  • In which skeletal muscles of Apoe-/- mice did you test DRP-1 and OPA1 of Figure 4?;

Quadriceps.

  • Is it possible to show separately the skeletal muscle weight? e. tibialis anterior, gastrocnemius exc.;

Due to an intrinsic limitation of the seahorse analysis and axonal excitability technique, we had to use the samples soon after the isolation. Quadriceps were used to extract proteins. Thus, we do not have this info.

  • Since the authors observed that atorvastatin influenced the fibers responsiveness might be interesting to analyze if this aspect should induce some modifications in the skeletal muscle fiber composition, thus an experiment to check the slow and fast myofibers should be added.

We agree with the reviewer, but for the reason previously mentioned we do not have the possibility to perform new experiments. Thus, this aspect was listed among the limitations as follows “Data of the present manuscript have to be interpreted in the frame of some limitations. First, the use of a wildtype model would have strengthened the results, although ApoE-/- is one of the most validated models of preclinical atherosclerosis [24]. Second, the mitochondrial respiration would need to be replicated in the single fibres, e.g., soleus, gas-trocnemius, or quadriceps. Third, skeletal muscle fibres composition could have been studied through a histological approach, namely, by using an immunofluorescence co-staining protocol that can simultaneously detect type I, IIa and IIb myosin fibres by means of BA-D5, SC-71 and BF-F3 specific antibodies [62]. Fourth, skeletal muscle composition could be determined by microgenomic approach, based on differences in the expression of genes in slow-oxidative (type 1) and fast-glycolytic (type 2B) fibres, through transcriptome analysis at the single fibre level (microgenomics) [63].”

Round 2

Reviewer 3 Report

In the revised version of the article “impact of atorvastatin on skeletal muscle mitochondrial activity, locomotion and axonal exitability – evidence from Apoe-/- mice”, the authors change all the minor points observed in the previous version and include in the section “conclusions” the limitations of this study, since they could not perform new in vivo experiments. Thus, the current version of the manuscript in my opinion is suitable for the publication in International Journal of Molecular Sciences.